# Characterization and In Vitro Fecal Microbiota Regulatory Activity of a Low-Molecular-Weight Exopolysaccharide Produced by *Lactiplantibacillus plantarum* NMGL2

**DOI:** 10.3390/foods11030393

**Published:** 2022-01-29

**Authors:** Mengke Yao, Min Zhang, Tiantian Lai, Zhennai Yang

**Affiliations:** Beijing Advanced Innovation Center for Food Nutrition and Human Health, Beijing Engineering and Technology Research Center of Food Additives, Beijing Technology and Business University, No. 11 Fu-Cheng Road, Hai-Dian District, Beijing 100048, China; 1930201080@st.btbu.edu.cn (M.Y.); 1850201008@st.btbu.edu.cn (M.Z.); 1930201070@st.btbu.edu.cn (T.L.)

**Keywords:** *Lactiplantibacillus plantarum* NMGL2, exopolysaccharide, characterization, fecal microbiota

## Abstract

The exopolysaccharide (EPS) produced by *Lactiplantibacillus plantarum* NMGL2 isolated from traditional fermented dairy cheese was purified chromatographically with DEAE-Sepharose and Sepharose CL-6B columns. The purified EPS was characterized by various physicochemical methods and in vitro fecal microbiota regulation assay. The results showed that the EPS had a relatively low molecular weight of 3.03 × 10^4^ Da, and it had a relatively high degradation temperature of 245 °C as determined by differential scanning calorimetry. Observation of the EPS by scanning electron microscopy, transmission electron microscopy, and atomic force microscopy revealed a highly branched and tangled fibrous network microstructure with many hollow microtubules and spherical particles. Structural study by 1H NMR spectroscopy suggested that the EPS contained a tetrasaccharide repeating unit with monosaccharide components of β-galactose (4.6%), α-glucose (20.6%), and α-mannose (74.8%). The EPS was highly resistant to hydrolysis of simulated human saliva, gastric, and intestinal juices. Moreover, the EPS beneficially affected the composition and diversity of the fecal microbiota, e.g., increasing the relative abundance of *Firmicutes* and inhibiting that of *Proteobacteria*. The results of this study indicated significant bioactivity of this novel low-molecular-weight EPS produced by *Lpb. plantarum* NMGL2, which could serve as a bioactive agent for potential applications in the food and health care industry.

## 1. Introduction

Many species of lactic acid bacteria (LAB) can synthesize exopolysaccharides (EPSs), including capsular polysaccharides (CPS) and slime polysaccharides (SPS) [1], such as *Streptococcus thermophilus*, *Lactobacillus acidophilus*, *Lacticaseibacillus rhamnosus*, *L**actiplantibacillus*
*plantarum*, *Lacticaseibacillus casei*, and *Lactococcus lactis* [2]. Depending on the composition of the polysaccharides, the EPSs of LAB can be classified into homopolysaccharides and heteropolysaccharides. Structurally, the repeating units of the EPSs from LAB might be linear or highly branched, with the molecular weight (Mw) of the EPSs up to about 10^6^ Da [3]. These EPSs have been shown to possess various bioactivities, such as antitumor [4], immunomodulation, interfering with adhesion of pathogens, inhibiting proliferation of the human colon cancer cell line, and improving resistance against viral infections [5].

The functional properties of the EPSs of LAB might vary greatly with the producing microorganisms, their culture conditions, and media composition, as well as the Mw of the polysaccharides. The EPSs with high Mw had great potential in improving food structure, e.g., promoting the formation of EPS-protein network structure in the fermentation process [6], limiting the shrinkage sensitivity of fermented milk. High Mw EPSs were also shown to help microbial communities to tolerate extreme temperatures and provide better protection against extreme cold temperatures [7]. Meanwhile, EPSs of lower Mw had higher water solubility with relatively extended chain conformation leading to better bioactivity [8], including immunomodulatory, antioxidant, and antitumor activities [9]. The lower Mw EPSs might also be used more efficiently by intestinal microbes [10]. It was shown that bacterial EPS364 with a relatively lower Mw of 1.48 × 10^4^ Da was a promising antitumor drug with obvious antitumor effects by inhibiting cancer cell growth and adhesion [11].

The human intestinal microbiota is a diverse microbial community dominated by bacteria, and maintenance of the dynamic microbiota balance is of importance in the prevention and treatment of many diseases. It was reported that the EPSs produced by *Lactobacillus delbrueckii* ssp. *bulgaricus* SRFM-1 [12] and *Lactococcus lactis* [13] were indigestible in the human body and could reach the end of the human intestinal tract to exert their regulatory function on the intestinal microbiota. EPSs of LAB could regulate the intestinal microbiota mainly by improving the adhesion of microbes to the intestinal mucosa, providing energy for microbial growth, balancing intestinal microbiota, and improving intestinal immunity [14]. Although the potential biological functions of the EPSs from LAB were confirmed, evaluation of these activities was mainly focused on the impact on individual beneficial bacterial species (*Bifidobacteria*, *Lactobacill**us*, etc.), and the impact on the whole intestinal microbiota was not well studied.

In this study, we reported a novel low-molecular-weight EPS produced by *Lpb. plantarum* NMGL2 that was purified chromatographically and characterized physicochemically and biologically. 16S rRNA amplification and sequencing analysis were carried out to explore the composition, abundance, and structural changes of the fecal microbiota, including beneficial and harmful bacteria under the intervention of the EPS. Meanwhile, the changes of short-chain fatty acids (SCFAs) during in vitro fermentation were determined. The present study would help researchers better understand the physicochemical properties and bioactivity of the low-molecular-weight EPS produced by *Lpb. plantarum* NMGL2 for potential applications in functional foods.

## 2. Materials and Methods

### 2.1. Bacterial Strain and Culture Condition

*Lpb. plantarum* NMGL2 was isolated from traditional fermented dairy cheese in Inner Mongolia of China and stored at −80 °C in Dairy Laboratory in Beijing Technology and Business University of China. Three consecutive transfers were performed at 37 °C for 18 h in MRS medium (Beijing Aoboxing Co., Ltd., Beijing, China) to activate the strain from the freezing stock (−80 °C). The skim milk medium (10%) prepared by reconstitution of skim milk powder (Fonterra, Auckland, New Zealand) was used for the strains growth to prepare EPS samples in this study.

### 2.2. Extraction and Purification of EPS

The activated *Lpb. plantarum* NMGL2 was inoculated (3% *v*/*v*) into liquid skim milk medium and cultured anaerobically at 37 °C for 24 h to produce EPS. Then the enzymes were inactivated by heating the culture in a boiling water bath for 15 min. The 80% (*w*/*v*) trichloroacetic acid was added to the culture, and the final concentration was adjusted to 4% (*w*/*v*), which was stirred for 2 h before centrifuging at 11,000× *g* at 4 °C for 45 min. The supernatant without free cells and proteins was mixed with two volumes of anhydrous ethanol, stored at 4 °C for 12 h, and centrifuged at 11,000× *g* at 4 °C for 30 min. The precipitate was dialyzed and then lyophilized to obtain the crude EPS.

The crude EPS solution (20 mg/mL, 5 mL) was subjected to a DEAE-Sepharose Fast Flow anion-exchange chromatography column (26 × 400 mm). During stepwise gradient elution, distilled water was used for collection of initial 1–30 tubes, 0.2 mol/L NaCl solution used for subsequent 31–70 tubes, and finally 0.5 mol/L NaCl solution used for 71–100 tubes. Each tube of 5 mL of eluent was collected automatically in 5 min, and the carbohydrate content was determined by the phenol-sulfuric acid method. The eluate containing EPS was collected, dialyzed, lyophilized, and then further purified with a Sepharose CL-6B column (25 × 500 mm) to obtain the purified EPS.

### 2.3. Fourier Transform Infrared Spectroscopy Analysis

FT-IR spectroscopy was used to identify the major functional groups of the EPS. The mixture of purified EPS (2 mg) and KBr (200 mg) was ground to prepare the sample pellet. The FT-IR spectra were recorded in the region of 4000 cm^−1^ to 400 cm^−1^.

### 2.4. Molecular Weight Determination

The gel permeation chromatography (GPC) system including Shodex SB-806m-HQ column (13 μm, 300 × 8.0 mm) linked to an SB-G (10 μm, 50 × 6.0 mm) guard column was used to determine the molecular weight of EPS. The EPS was detected using a refractive index detector (RI) (Optilab Wyatt, Wyatt Technology, Santa Barbara, CA, USA) and a multi-angle laser-light scattering detector (MALLS) (DAWN HELEOS-II Wyatt, Wyatt Technology, Santa Barbara, CA, USA). The EPS sample (200 μL) was loaded to the column, which was eluted with 0.1 M NaNO_3_ solution at a flow rate of 0.5 mL/min. Data were processed with Wyatt Astra software (Version 5.3.4.14, Wyatt Technology, Santa Barbara, CA, USA).

### 2.5. Thermodynamic Stability Analysis

Differential scanning calorimetry (DSC) and thermal gravimetric analysis (TGA) were performed by a dta-dsc thermal analyzer. The EPS sample (5 mg) was placed into an Al_2_O_3_ crucible. The experiments were conducted from 20 to 600 °C at a heating rate of 10 °C/min. The weight loss and heat flow curve relative to temperature were used to generate the TG-DSC thermogram.

### 2.6. Microstructural Analysis of EPS

#### 2.6.1. Scanning Electron Microscopy Analysis

The purified EPS sample (5 mg) was fixed to the mica surface and coated with a gold layer on both sides. Then, the sample was observed using a scanning electron microscope (S-4800; Hitachi Ltd., Tokyo, Japan).

#### 2.6.2. Atomic Force Microscopy Analysis

The purified EPS was added to distilled water and stirred at 40 °C for 2 h to prepare EPS solution (1 mg/mL), which was then diluted to 10 μg/mL. A total of 5 μL of diluted EPS solution was dropped on the surface and dried at 25 °C. The Dimension Icon microscope (Bruker Instruments Co., Karlsruhe, Germany) was used to observe AFM images.

#### 2.6.3. Transmission Electron Microscopy Analysis

A drop of EPS sample (0.1 mg/mL) was placed on a carbon copper grid for 15 min, then stained with phosphotungstate for 3 min and dried completely with a vacuum desiccator. The transmission electron microscope was used to obtain images.

### 2.7. Monosaccharide Composition Analysis

To determine monosaccharide composition, the purified EPS sample (5 mg) was hydrolyzed with 2 M trifluoroacetic acid (TFA) at 120 °C for 3 h. After the removal of water and TFA from the sample, the methanol was added to the dry sample and evaporated under reduced pressure, and repeated five times. Then, the sample was reduced with 30 mg of NaBH_4_, and 1 mL of acetic anhydride was added to the hydrolyzed sample for acetylation. The Agilent Technologies 7890A GC with a DB-5 capillary column (30 × 0.25 mm) and flame ionization detector was used to examine the acetylated sample. The carrier gas was N_2_, with a flow rate of 1 mL/min. The monosaccharide composition was identified by comparing them to standard L-fucose, L-rhamnose, L-arabinose, D-galactose, D-glucose, D-xylose, D-mannose, D-fructose, and D-ribose.

### 2.8. Nuclear Magnetic Resonance Spectroscopy Analysis

The purified EPS sample was first dissolved in D_2_O, then freeze-dried, and this procedure was repeated twice. The sample was then redissolved in D_2_O (5 mg/mL) and ready for ^1^H NMR testing. The NMR spectrum was obtained with a Bruker AVANCE AV-600MHz spectrometer (Bruker Group, Billerica, MA, USA). Parts per million (ppm) was used to express the chemical shift.

### 2.9. In Vitro Digestibility of EPS

The saliva digestion was carried out using the previous method [15]. The EPS sample (10.0 mg/mL) was removed from a 37 °C water bath after 2 min incubation, and the enzyme was inactivated by heating the incubation system in a boiling water bath. Preparation of gastric juice (pH 1.5) was performed with reference to a previous method [16]. Hydrolysis of EPS by gastric juice was performed as follows with the final concentration of *Lpb. plantarum* NMGL2 EPS at 10.0 mg/mL. The samples were taken out at regular intervals from a 37 °C water bath and placed in a boiling water bath for 5 min. The simulated small intestinal juice was prepared with reference to a previous method [17]. The samples added with EPS were drawn at regular time intervals from a 37 °C water bath, and then the enzymes were inactivated. The reducing sugar content and total content of the sample were determined by the DNS method and phenol-sulfuric acid method, respectively. The percent hydrolysis was calculated as: Hydrolysis (%) = [Reducing sugar released/(Total sugar − Initial reducing sugar)] × 100.

### 2.10. Fecal Microbiota Regulatory Activity of EPS

#### 2.10.1. Fecal Slurry Preparation

Preparation of fecal slurry was performed by using a previously described method with slight modifications [18]. The fresh fecal samples were provided by three healthy volunteers (ages 20–28) who had no history of gastrointestinal disease and were not treated with antibiotics within six months. The phosphate buffer solution (PBS, 1.0 mol/L, pH 7.0) was added to the mixture of fecal samples to prepare fecal diluent in a sterile environment. The fecal diluent was homogenized for 2 min and centrifuged at 30× *g* for 5 min to remove the large particles. Then, the leaving liquid was used as the inoculum for the fecal fermentation.

#### 2.10.2. In Vitro Fecal Fermentation

The in vitro fecal fermentation was performed by using a previous method with slight modifications [19]. Briefly, the basal culture medium was formulated as: peptone water 2 g/L, yeast extract 2 g/L, NaCl 0.1 g/L, K_2_HPO_4_ 0.04 g/L, KH_2_PO_4_ 0.04 g/L, MgSO_4_·7H_2_O 0.01 g/L, CaCl_2_ H_2_O 0.01 g/L, NaHCO_3_ 2 g/L, bile salts 0.5 g/L, cysteine-HCl 0.5 g/L, Tween 80 2 g/L; pH 7.0. The medium was autoclaved (121 °C, 15 min), and then 1 mL/L of hemin solution (50 mg/mL) and 10 mL/L of vitamin K previously sterilized by filtration were added. The purified EPS sample (200 mg) was added into a 10 mL basal culture medium and fully dissolved by oscillation to obtain a culture system with an EPS concentration of 20 mg/mL. A total of 1.5 mL of fecal slurry was added to each culture system, and the incubation was performed inside an anaerobic cabinet (Model Plas Labs 855-AC; PLAS LABS, Inc., Lansing, MI, USA) at 37 °C. The liquid samples were taken after 4, 12 h of fermentation. The above experimental procedure without EPS was performed simultaneously as a blank control. These experiments were repeated five times.

#### 2.10.3. Extraction of Genomic DNA and Amplicon Sequencing of 16S rRNA Gene

According to the previous method [20], 16S rRNA gene analysis was performed to examine the diversity of fecal microbiota. The culture system was centrifuged at 200× *g* for 5 min to remove the large particle precipitation and then centrifuged at 9000× *g* for 5 min to collect the bacteria. Microbial community genomic DNA was extracted using the FastDNA^®^ (Spin Kit for Soil, MP Biomedicals, Santa Ana, CA, USA). The ABI GeneAmp^®^ 9700 PCR thermocycler (ABI, Foster City, CA, USA) was used to amplify the hypervariable region V3-V4 of the bacterial 16S rRNA gene with primer pairs 338F (5′-ACTCCTACGGGAGGCAGCAG-3′) and 806R (5′-GGACTACHVGGGTWTCTAAT-3′). Purified amplicons were pooled in equimolar and paired-end sequenced on an Illumina MiSeq PE300 platform (Illumina, San Diego, CA, USA) according to the standard protocols by Majorbio Bio-Pharm Technology Co., Ltd. (Shanghai, China). UPARSE version 7.1 was used to cluster operational taxonomic units (OTUs) with a 97% similarity, and chimeric sequences were discovered and removed. The RDP Classifier version 2.2 was used to compare the taxonomy of each OTU representative sequence to the 16S rRNA database (e.g., Silva v138) using a confidence threshold of 0.7.

#### 2.10.4. Determination of Short-Chain Fatty Acids

Short-chain fatty acids in the fecal cultures were determined by the 7890–5977 GC-MS system (Agilent Technologies Inc. Santa Clara, CA, USA). The experiment was performed according to the previous method [21] with slight modifications. Briefly, the culture was centrifuged at 9000× *g* for 5 min to collect the supernatant, which was added to NaOH (5 mmol/L) to extract SCFAs. Then one-step derivatization was carried out using propyl chloroformate in a reaction system of water, propanol, and pyridine by two-step extraction with n-hexane. The sample was injected into GC-MS, which was equipped with an HP-5 (30 m × 0.25 mm × 0.25 µm) column using helium as the carrier gas at a flow rate of 1.0 mL/min. The oven temperature was maintained at 70 °C for 5 min and then adjusted to 100 °C at the rate of 6 °C/min. The selection ionization mode (SIM) was selected as the detector operating mode. Six SCFA standards (Aladdin^®^, Shanghai, China) were used for identification and quantification, including acetic acid, propionic acid, n-butyric acid, i-butyric acid, n-valeric acid, and i-valeric acid.

### 2.11. Statistical Analysis

Data were expressed as mean ± SD, and the statistical significance of the difference was tested by one-way ANOVA and Student’s *t*-test by using SPSS 18.0 software. *p* values < 0.05, 0.01, 0.001 are indicated by *, **, ***, respectively.

## 3. Results and Discussion

### 3.1. Extraction and Purification of EPS

The yield of crude EPS produced by *Lpb. plantarum* NMGL2 was 380 mg/L. The crude EPS was subjected to purification through a DEAE-Sepharose Fast Flow column to obtain one major peak of EPS and a small peak upon gradient elution (Figure 1A). Subsequent purification of the EPS by Sepharose CL-6B gel permeation chromatography (Figure 1B) showed a single peak. The purified EPS was collected for further analyses.

### 3.2. Infrared Spectrum Analysis of EPS

The FT-IR absorption spectrum of EPS was shown in Figure 2, which exhibited a variety of typical absorption peaks of polysaccharides in the range of 4000–400 cm^−1^. A broad and strong stretching peak at 3283.25 cm^−1^ indicated the presence of a significant number of hydroxyl groups (O-H), which confirmed that the compound was a polysaccharide [22]. The C-H stretching vibration led to a weak peak at 2930.81 cm^−1^ [23]. The absorption at 1643.36 cm^−1^ was due to the stretching vibration of the C = O bond and carboxyl groups [24]. The weak stretch band at 812.26 cm^−1^ indicated the presence of α-d-glucose [25]. A broad stretch of C-O-C and C-O-H at 1000–1200 cm^−1^, and the absorption at 1019.66 cm^−1^ further confirmed that the polymer was a polysaccharide [26].

### 3.3. Molecular Weight Determination of EPS

The molecular weight of the EPS from *Lpb. plantarum* NMGL2 was determined to be 3.03 × 10^4^ Da by the GPC-RID-20 system, which was similar to the EPS produced by *Lpb. plantarum* KX041 (3.867 × 10^4^ Da) [27], but lower than the EPS from *Lpb. plantarum* DM5 [28] (1.11 × 10^6^ Da). Different bacterial sources, culture conditions, and hereditary characteristics may lead to differences in molecular weights.

As shown in the chromatogram (Figure 3), a single and symmetrical peak was observed with the Mw/Mn ratio of the EPS of 2.71, confirming that the *Lpb. plantarum* NMGL2 EPS was homogeneous. The bioactivity of EPS was reported to be affected by its molecular weight, and the EPS with low molecular weight seemed to be more effective for immunomodulatory activity and antitumor activity when compared with the high molecular weight EPS [8].The EPS from *Lpb. plantarum* NMGL2 with a relatively lower molecular weight suggested the significance of this polysaccharide for potential application as a bioactive agent.

### 3.4. Thermogram Analysis of EPS

TGA determines the dynamic weight loss of the purified EPS produced by *Lpb. plantarum* NMGL2. As shown in Figure 4, the heating process of the EPS included two steps. In the first step, an initial weight loss of approximately 10.59% was observed, which mainly attributed to the water loss in the EPS sample from 20 to 175 °C. This moisture loss of mass indicated that the EPS had a high abundance of carboxyl groups, which were bound to a greater number of water molecules [29]. In the second step, there was an increased weight loss of the sample up to 50.67% at around 400 °C, and about 19.08% of the total EPS was not degraded. When the EPS was heated to a high temperature, the C-C and C-O bonds in the ring structure were broken, leading to the release of CO, CO_2_, and H_2_O [30]. The EPS started to degrade at about 245 °C, which was consistent with the fastest weight loss temperature. This indicated that the EPS had high thermal stability and had a broad application prospect in the food industry.

### 3.5. Microstructural Analysis of EPS

#### 3.5.1. SEM Analysis

Scanning electron microscopy was used to study the surface morphology of macromolecules that is associated with their physical properties. As shown in Figure 5 under 1000× (Figure 5A) and 5000× (Figure 5B) magnifications, the purified EPS produced by *Lpb. plantarum* NMGL2 appeared to be a highly branched network structure with many hollow microtubules, indicating the EPS with strong water holding capacity and viscous nature that might be potentially applied in the improvement of the physical properties of foods. The EPS from *Lpb. fermentum* Lf2 could increase the hardness and consistency index, decrease the flow behavior index of yogurt and produce yogurt with no sensory defects [31]. In addition, EPS could replace hydrocolloids to improve the volume, texture, and shelf life of bread [32]. A similar reticulated porous appearance was also found with the purified EPS of *Lysinibacillus fusiformis* KMNTT-10 [33].

#### 3.5.2. AFM Analysis

Atomic force microscopy provides an effective approach to investigating the three-dimensional microstructure of polymers. The AFM images of *Lpb. plantarum* NMGL2 EPS were shown in Figure 6A,B. There was the presence of many fiber structures with heights ranging from 0.4 to 7.6 nm, which were significantly higher than those previously reported values (ca. 0.2–0.6 nm) [34]. This suggested that the network structures of the EPS from *Lpb. plantarum* NMGL2 might be due to the lateral association of the polymer’s multimolecular chains in varying degrees. As revealed in the AFM images of the EPS (Figure 6A,B), some regions formed fibrous networks, while other regions were relatively sparse, suggesting that the EPS structure might be tangled networks. This further confirmed the complex network structure of the EPS from *Lpb. plantarum* NMGL2, which might lead to a polymer with higher hydrophilicity and potential to be used in food production.

#### 3.5.3. TEM Analysis

Transmission electron microscopy is usually used to observe and analyze the ultrastructure of samples with high resolution and magnification. The TEM images of the purified EPS from *Lpb. plantarum* NMGL2 were presented in Figure 7A,B, demonstrating dendrimer-like microstructure that was highly branched with the presence of many spherical particles and polymer aggregates. At 1000× magnification, it could be observed that the EPS chains consisted of multiple strands of finer fibrous structures, with the smallest fibers less than 1 nm in diameter. The branching chains were connected with each other to form the dense structure of the polymer, which was in agreement with the observation of the EPS by SEM and AFM. Similar branched-chain morphology of the EPS from *Antrodia cinnamomea* as observed by TEM was reported earlier, with more flexible interchain entanglement after the addition of surfactant [35]. The microstructure of EPS was closely related to its physical properties, e.g., water binding capacity, which played an important role in maintaining the stable texture of fermented milk.

### 3.6. Monosaccharide Composition and NMR Spectroscopy of EPS

#### 3.6.1. Monosaccharide Analysis

Figure 8A,B revealed the presence of galactose, glucose, and mannose in the EPS produced by *Lpb. plantarum* NMGL2, indicating that the EPS was a heteropolymer. However, the ratio of each sugar was found to be varying in percentage, such as galactose (4.6%), glucose (20.6%), and mannose (74.8%). A similar kind of monosaccharide composition was observed in the EPSs of *Lpb. plantarum* WLPL09 [36]. However, the EPS produced by *Chaetomium globosum* CGMCC 6882 was composed of more sugars such as rhamnose, arabinose, galactose, glucose, xylose, mannose, galacturonic acid, and glucuronic acid [37].

#### 3.6.2. NMR Spectroscopy

^1^H NMR spectroscopy provides structural information of the monosaccharide components in the repeating unit of the bacterial EPSs by analyzing spectrum signals between 4.5 and 5.5 ppm to distinguish the anomeric protons of sugar residues. The ^1^H NMR spectrum of the EPS from *Lpb. plantarum* NMGL2 was presented in Figure 9, showing four major signals in the anomeric region at 5.22, 5.06, 5.01, and 4.97 ppm. This suggested that the EPS of *Lpb. plantarum* NMGL2 might be composed of a tetrasaccharide repeating unit. These resonance signals might be assigned to the anomeric protons of α-d-glucose (5.22), α-d-mannose (5.06), β-d-galactose (5.01), and α-d-glucose (4.97) according to the carbohydrate research database (www.glyco.ac.ru, accessed on 25 August 2021), consisting with the results of monosaccharide analysis described above. The detailed structure of the EPS will be characterized in another study.

### 3.7. In Vitro Digestibility of EPS

For a substrate to be a qualified prebiotic, it should resist GIT conditions. The EPS produced from *Lpb. plantarum* NMGL2 was shown with high resistance to simulated saliva, gastric juice, and intestinal juice. After 2 min of saliva digestion, the hydrolysis observed for the EPS was 0.54%. The effect of simulated human gastric juice and intestinal juice on hydrolysis of *Lpb. plantarum* NMGL2 EPS was shown in Figure 10A,B. Hydrolysis of *Lpb. plantarum* NMGL2 EPS in the gastric juice increased with the incubation time. The maximum hydrolysis was 1.26% at pH 1.5 after 6 h. The highest hydrolysis of EPS under simulated intestinal conditions was 2.67%. These results revealed that the EPS produced from *Lpb. plantarum* NMGL2 could resist human digestive enzyme hydrolysis and reach the colon as a substrate for fermentation by intestinal microbiota.

### 3.8. Fecal Microbiota Regulatory Activity of EPS

To determine the effect of *Lpb. plantarum* NMGL2 EPS on the composition of human fecal microbiota, a total of 1,264,293 sequences were obtained from 20 samples. Each sample was subsampled to the same sequencing depth (36,997 reads per sample), and clustered, 432 OTUs were obtained at 97% identity. The rarefaction curves (Figure 11A,B) for all samples reached a plateau, indicating that the sequencing depth was sufficient. The common and unique OTUs between groups were demonstrated by Venn diagram (Figure 11E). After incubation for 4 h, the number of OTUs in the control group and EPS group were 282 and 345, respectively, and the EPS significantly increased the number of unique OTU. After incubation for 12 h, a total of 343 and 266 OTU species were found in the control group and EPS group, respectively. Principle component analysis (PCA) revealed that different fecal coculture treatments caused unique structural changes of fecal microbiota, which was shown by clustering of samples in plots (Figure 11F). The first (PC1) and second (PC2) axes contributed 29.14% and 15.28% of the variation, respectively.

#### 3.8.1. α-Diversity Analysis

Differences in fecal microbiota before and after EPS intervention were determined using α-diversity (richness estimates and diversity values). Chao1 index was used to estimate richness, and Shannon index was used to estimate diversity values. After in vitro incubation for 4 h, the Chao1 index of the EPS group was significantly higher than the control group (*p* < 0.01) (Figure 11C), but decreased Shannon index compared with the control group (Figure 11D). After incubation for 12 h, the Chao1 index of the EPS group decreased compared with the control group, but the differences of Shannon index between the EPS group and the control group were not statistically significant.

#### 3.8.2. OTUs Analysis

The differences of fecal microbiota in different groups at phylum and genus levels were analyzed (Figure 12A,B) to further evaluate the influence of EPS on fecal microbiota. At the phylum level, the fecal microbiota was composed of *Firmicutes*, *Bacteroidetes*, *Actinobacteria*, and *Proteobacteria* (>99.5%) in both the samples incubated for 4 and 12 h. After incubation for 4 h, the relative abundance of *Firmicutes* increased by 42.14%, and *Bacteroidetes* decreased by 80% in the EPS group (EPS4) compared with the control group (control4). After 12 h incubation of the EPS group (EPS12), the relative abundance of *Actinobacteria* and *Bacteroidetes* increased by 64.48% and 56%, respectively, while *Proteobacteria* decreased by 66.05%, compared with the control group (control12). It was reported that inflammatory bowel diseases (IBD) could cause a decreased abundance of *Firmicutes* and increased abundance of *Proteobacteria* in the intestinal tract [38]. *Proteobacteria* was considered to be the microbial marker for dysregulation of the gut microbiota associated with many diseases [39]. The effect of the EPS produced by *Lpb. plantarum* NMGL2 on the fecal microbiota change could be due to the relatively low molecular weight (3.03 × 10^4^ Da) and monosaccharide composition of the EPS, which benefited digestion of the polymer by the microbiota and growth of certain microbes, e.g., *Firmicutes* in the early fermentation stage [18]. The EPS-induced increase in the abundance of *Actinobacteria* containing probiotic *Bifidoba**cterium* would be beneficial to human health. *Bifidobacterium* played an important role in maintaining the structural integrity of intestinal mucosa, regulating the inflammatory cytokines, and preventing the passage of pathogenic bacteria and toxins [40].

At the genus level, the healthy human gut was dominant with an average relative abundance of more than 5% of *Bifidobacterium*, *Ruminococcus*, *Faecalibacterium*, *Phascolarctobacterium*, *Blautia*, *Subdoligranulum*, *Bacteroides*, *Lactobacillus*, and *unclassified_f_Lachnospiraceae*. In vitro fecal fermentation with the EPS for 4 h the abundance of *Phascolarctobacterium*, *unclassified_f__Lachnospiraceae*, *Ruminococcus*, *Faecalibacterium*, *Roseburia* increased significantly (*p* < 0.001), and *Bacteroides*, *Subdoligranulum*, *Fusicatenibacter*, *Blautia*, *Alistipes* decreased significantly (*p* < 0.001) compared with the control group. The SCFAs were mainly produced by *Firmicutes* and *Actinobacteria*, and there was a significant increase in SCFAs producing genera in the EPS group after 4 h incubation, as indicated above. For instance, *Phascolarctobacterium* was an important producer of acetic acid and propionic acid [41], and *Faecalibacterium* was the main butyric acid producer that could degrade polysaccharides and had anti-inflammatory capability [42]. It was reported that propionic acid was mainly synthesized through the succinate, propanediol, and acrylate pathways [43]. There were two metabolic pathways for butyrate biosynthesis: one via butyryl-CoA/acetate CoA transferase, requiring a molecule of acetate; another one through the phosphotransbutyrylase-butyrate kinase pathway [44]. After incubation with the EPS for 12 h, the relative abundance of *Bifidobacterium* (*p* < 0.01), *Lactobacillus* (*p* < 0.001) and *Veillonella* (*p* < 0.001) increased significantly, but *Escherichia-Shigella*, *Lachnospiraceae_UCG-004*, *Lachnoclostridium* and *Parabacteroides* decreased significantly (*p* < 0.001). The changes of *Bifidobacterium* were consistent with *Actinobacteria*, and *Escherichia-Shigella* was related to *Proteobacteria*. It was reported that patients with irritable bowel syndrome had significantly decreased abundance of intestinal *Bifidobacteria*. *Lactobacillus* could produce bacteriocins and competitively exclude potential pathogens [45].

Further linear discriminant analysis (LDA) and effective size comparisons (LEfSe) were carried out to evaluate dominant fecal microbiota (LDA; values > 4) [46] as biomarkers between groups. After incubation for 4 h, our results showed that there were 11 and 9 significant differences in the EPS and control groups (EPS4, control4), respectively (Figure 13A). After incubation for 12 h, it demonstrated that there were 18 and 10 significant differences in the EPS and control groups (EPS12, control12), respectively (Figure 13B). To sum up, the results described above, the EPS produced by *Lpb. plantarum* NMGL2 could generally increase the relative abundance of beneficial bacteria and reduce the relative abundance of harmful bacteria in the fecal microbiota.

#### 3.8.3. SCFAs Produced in the Fecal Cultures

SCFAs include acetic, propionic, and butyric acids, which were produced by fermentation of dietary fiber and prebiotic carbohydrates by microflora. SCFAs could provide nutrition and energy for intestinal epithelial cells and play important physiological roles in the host. The increase in SCFAs is beneficial to the health of the host, such as suppressing pathogenic bacteria in the gut [47]. In addition, lactic acid produced by *Firmicutes* and *Actinobacteria* may play an important role in the regulation of *Proteobacteria*, especially *Escherichia-Shigella*. Therefore, the changes of SCFAs and lactic acid during in vitro fecal fermentation could reflect the bioactivity of *Lpb. plantarum* NMGL2 EPS. Changes of SCFAs concentrations in feces during in vitro fecal fermentation as affected by the EPS produced by *Lpb. plantarum* NMGL2 are shown in Figure 14. After EPS intervention for 4 h, though there was a significant increase (*p* < 0.001) in the abundance of SCFAs producing *Phascolarctobacterium* and *Faecalibacterium*, the levels of SCFAs decreased significantly, probably due to the fact that the SCFAs were used by *Firmicutes* to increase their growth. Furthermore, the contents of acetic acid and propionic acid increased significantly after 12 h. However, the concentration of the total SCFAs during the in vitro fecal fermentation was not significantly changed, probably due to the fact that the SCFAs products of some bacteria might provide substrates for others. This was also known as the cross-feeding effect that might contribute to the alteration of microbiota composition [48]. In addition, the concentration of butyrate decreased significantly compared to both control groups (4 and 12 h). It was reported that SCFAs as substrates, e.g., dietary butyrate, could significantly increase the relative abundance of the phylum *Firmicutes* in mice [49]. Lactic acid was also produced during EPS fermentation in vitro, as found in other studies [12]. Therefore, the EPS may affect the fecal SCFAs and lactic acid composition, which in turn could be as substrates to regulate the fecal microbiota composition. However, the exact mechanism of the EPS-induced complex cross-feeding effect involving SCFAs on fecal microbiota changes needs to be further studied.

## 4. Conclusions

In this study, a novel EPS with a low molecular weight (3.03 × 10^4^ Da) produced by *Lpb. plantarum* NMGL2 isolated from traditional cheese in Inner Mongolia of China was characterized with significant bioactivity by regulating the composition and diversity of fecal microbiota. The EPS produced by *Lpb. plantarum* NMGL2 had a strong resistance to human digestive juices (saliva, gastric juice, and intestinal juice). It was relatively heat stable with a degrading temperature of about 245 °C. Microstructurally, the EPS appeared to be a highly branched and tangled fibrous network with many hollow microtubules and spherical particles. Monosaccharide analysis indicated that the EPS was composed of β-galactose (4.6%), α-glucose (20.6%), and α-mannose (74.8%), and it might contain a tetrasaccharide repeating unit. Analysis of the fecal microbiota regulatory activity of the EPS showed that the EPS affected the composition and diversity of the fecal microbiota by increasing the relative abundance of beneficial bacteria, e.g., *Firmicutes*, and inhibiting harmful bacteria, e.g., *Proteobacteria.* The EPS also affected the fecal SCFAs composition during in vitro fecal fermentation, which in turn could regulate the fecal microbiota. Therefore, the low-molecular-weight EPS produced by *Lpb. plantarum* NMGL2 possessed significant bioactivity, which could be explored for potential applications in the food and health care industry.

## Figures and Tables

**Figure 1 foods-11-00393-f001:**
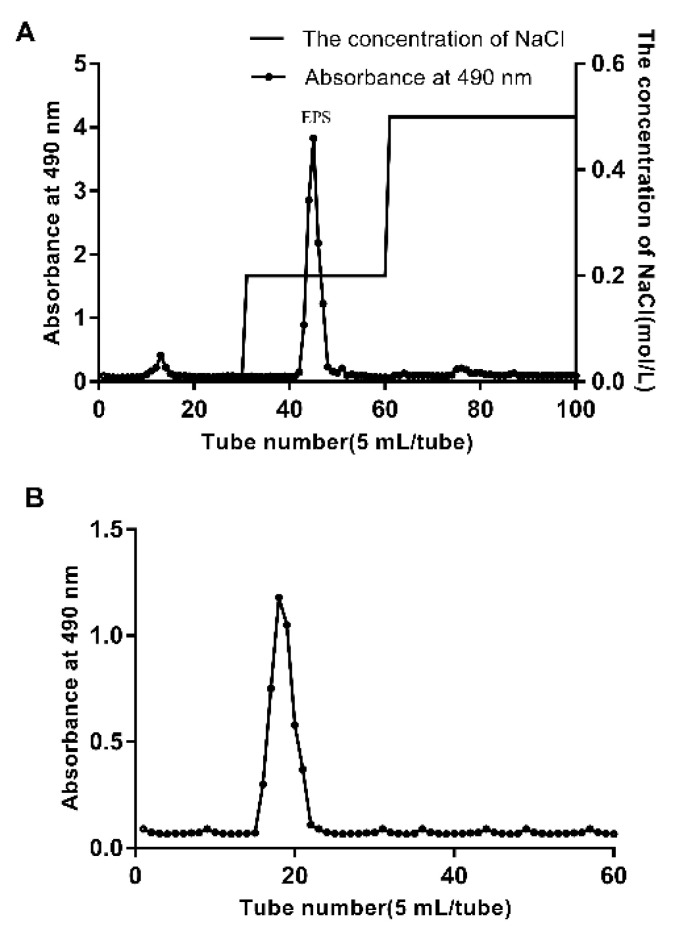
Purification of the crude EPS from *Lpb. plantarum* NMGL2 by anion-exchange chromatography on a DEAE-Sepharose Fast Flow column (**A**), and further gel permeation chromatography on a Sepharose CL-6B column (**B**).

**Figure 2 foods-11-00393-f002:**
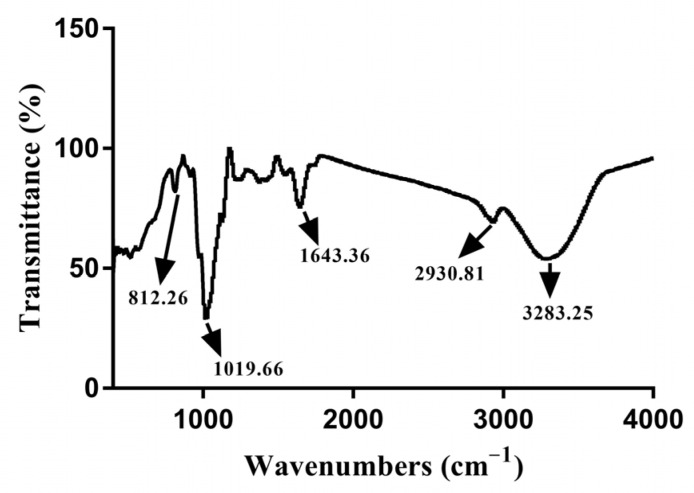
FT-IR spectrum of the EPS from *Lpb. plantarum* NMGL2 in the range of 4000–400 cm^−1^.

**Figure 3 foods-11-00393-f003:**
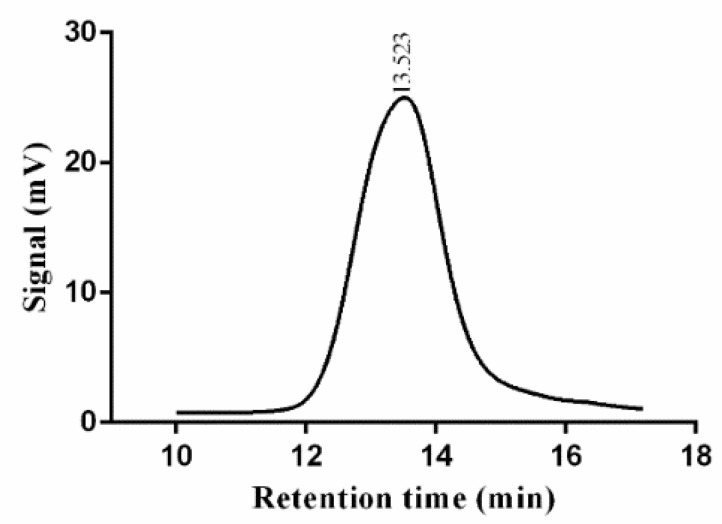
Molecular weight determination of the EPS from *Lpb. plantarum* NMGL2 by gel permeation chromatography.

**Figure 4 foods-11-00393-f004:**
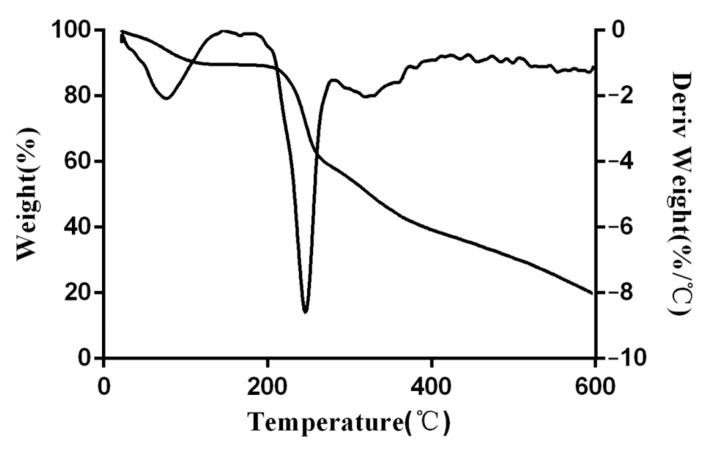
Thermogram curves of the EPS produced by *Lpb. plantarum* NMGL2.

**Figure 5 foods-11-00393-f005:**
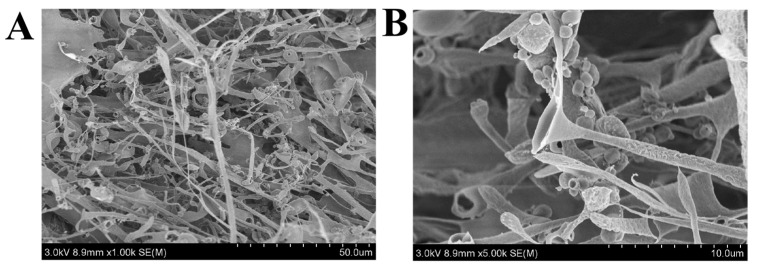
Scanning electron microscope images ((**A**): 1000×, (**B**): 5000×) of the EPS produced by *Lpb. plantarum* NMGL2.

**Figure 6 foods-11-00393-f006:**
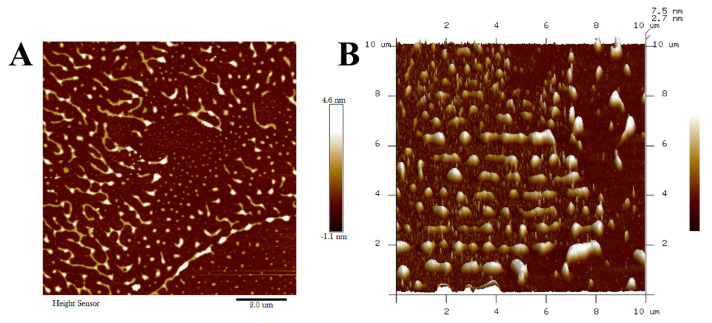
Atomic force microscopy images ((**A**): planar image; (**B**): cubic image) of the EPS produced by *Lpb. plantarum* NMGL2.

**Figure 7 foods-11-00393-f007:**
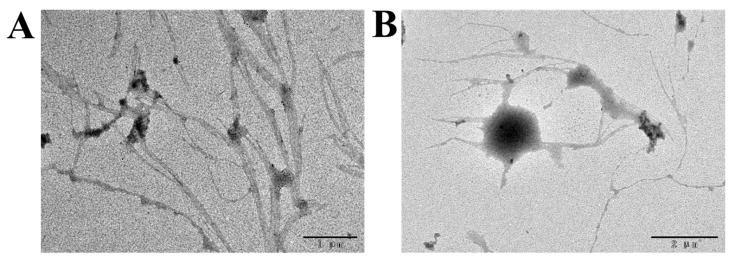
Transmission electron microscopic images ((**A**): 2000×, (**B**): 4000×) of the EPS produced by *Lpb. plantarum* NMGL2.

**Figure 8 foods-11-00393-f008:**
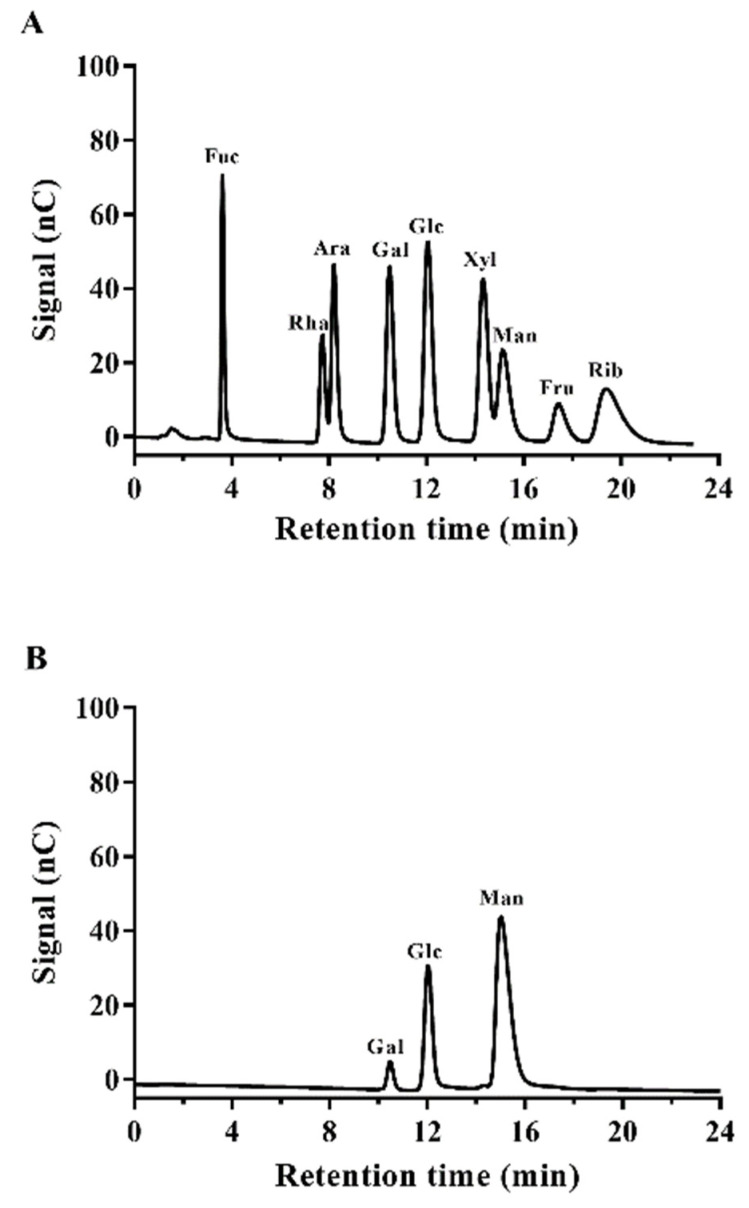
Gas chromatograms of the standard monosaccharides (**A**) and the EPS produced by *Lpb. plantarum* NMGL2 (**B**).

**Figure 9 foods-11-00393-f009:**
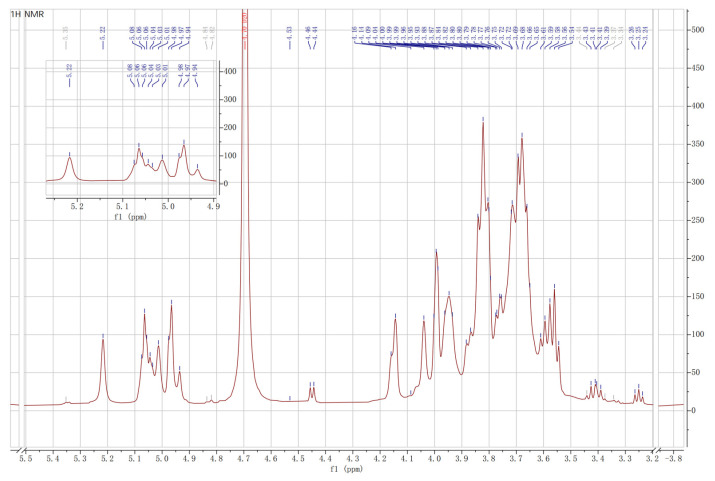
^1^H NMR spectroscopy of the EPS from *Lpb. plantarum* NMGL2.

**Figure 10 foods-11-00393-f010:**
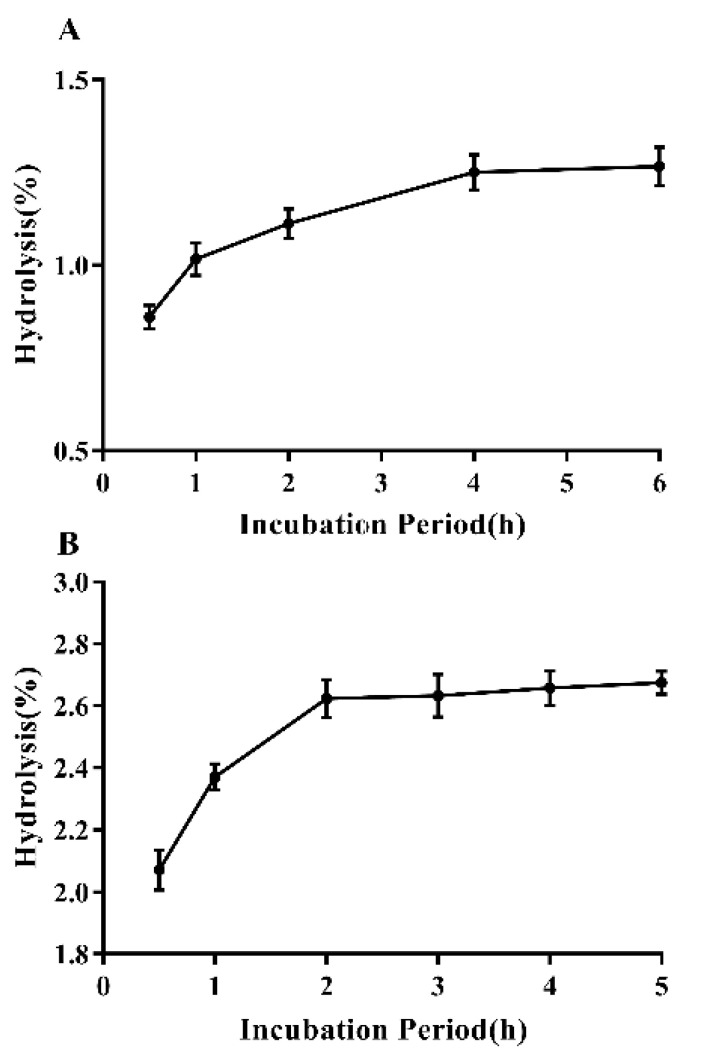
Hydrolysis of *Lpb. plantarum* NMGL2 EPS in simulated human digestive juice: (**A**). Gastric juice; (**B**). Intestinal juice.

**Figure 11 foods-11-00393-f011:**
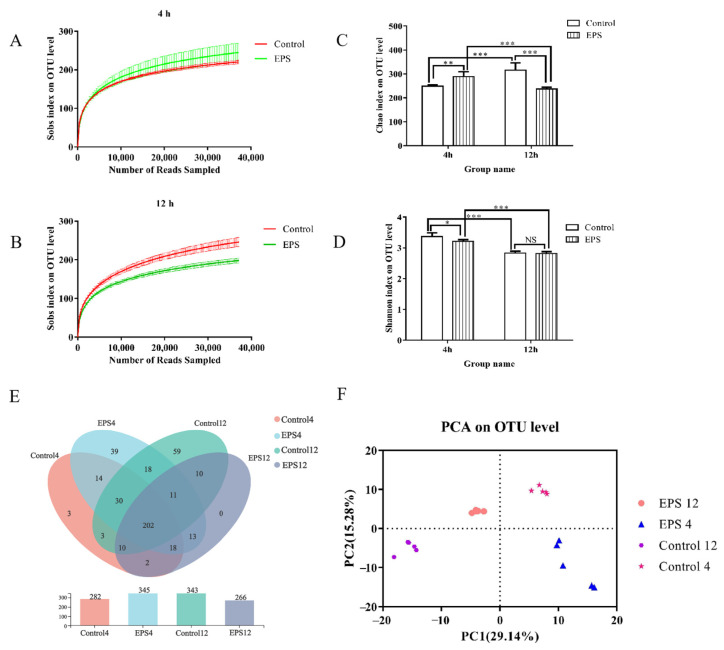
Analysis of DNA sequence data obtained from in vitro fecal fermentation under the intervention of the EPS produced by *Lpb. plantarum* NMGL2: Rarefaction curves for each experimental group with an incubation time of 4 h (**A**) and 12 h (**B**); α-diversity of fecal microbiota in terms of microbial richness estimates Chao1 (**C**) and diversity indices Shannon (**D**) measured at OTUs definition of >97% identity, *p* values < 0.05, 0.01, 0.001 are indicated by *, **, ***, respectively; Venn diagrams showing the unique and shared OTUs of different treatment samples (**E**); PCA analysis of fecal microbiota samples (**F**).

**Figure 12 foods-11-00393-f012:**
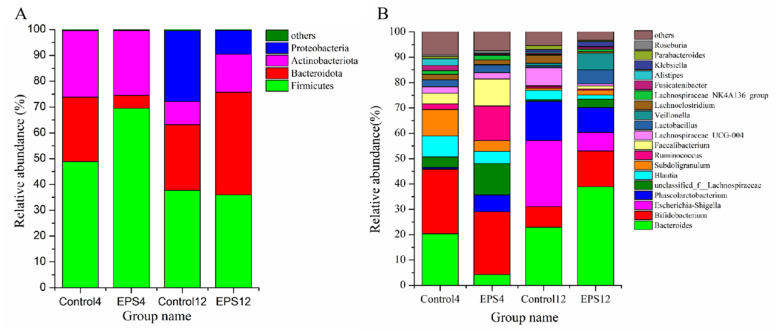
Comparison of fecal microbiota as affected by the EPS produced by *Lpb. plantarum* NMGL2 at the phylum level (**A**) and genus level (**B**) after incubation of the samples for 4 h (Control4 and EPS4 groups) and 12 h (Control12 and EPS12 groups).

**Figure 13 foods-11-00393-f013:**
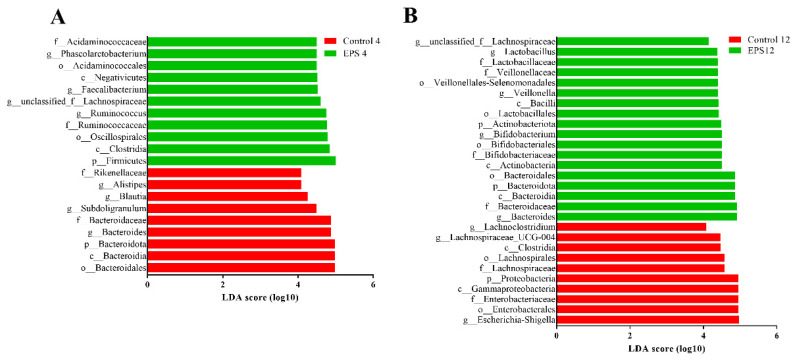
Comparison of fecal microbiota as affected by the EPS produced by *Lpb. plantarum* NMGL2 by linear discriminant analysis (LDA score > 4) generated by LEfSe analysis after incubation of the samples for 4 h (Control4 and EPS4 groups) (**A**) and 12 h (Control12 and EPS12 groups) (**B**).

**Figure 14 foods-11-00393-f014:**
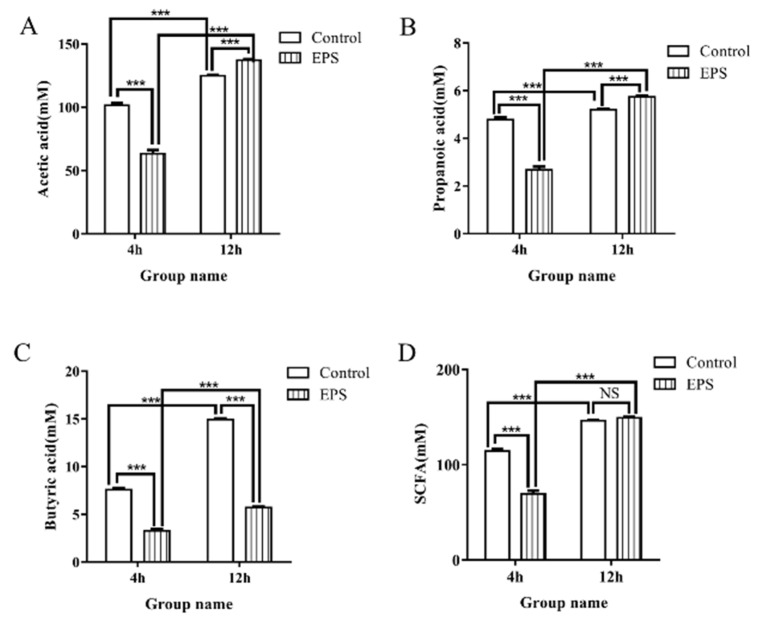
Mean concentrations of acetate (**A**), propionate (**B**), butyrate (**C**), and total short-chain fatty acids (SCFAs) (**D**) during in vitro fecal fermentation with intervention of the EPS produced by *Lpb. plantarum* NMGL2 for 4 h and 12 h. *p* values < 0.001 is indicated by ***. NS: no significant.

## Data Availability

Not applicable.

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
