# Peer review of "Characterization and In Vitro Fecal Microbiota Regulatory Activity of a Low-Molecular-Weight Exopolysaccharide Produced by Lactiplantibacillus plantarum NMGL2"

_foods, 2022, doi:10.3390/foods11030393_

Round 1

Reviewer 1 Report

The study "Characterization and in vitro intestinal microflora regulatory activity of a low-molecular-weight exopolysaccharide produced by Lactobacillus plantarum NMGL2" purposed by Yao and co-workers aimed to characterize by the application of multiple analytic methods the novel low-molecular-weight exopolysaccharide (EPS) produced by L. plantarum NMGL2. The scientific soundness of this work is of relevance in accordance with the perspective application of this EPS as prebiotic source.

The first sections, ones related to chemical and structural characterization of this metabolite, did not require further revisions.

Oppositely, the microbiological field of investigation need to be significantly revised and discussed. Berg and co-authors have recently revised the correct terminology to be applied (PMID: 32605663). Therefore, all flora or microflora terms used along the text need to be changed in microbiota accordance to the use of appropriate terminology. Applying an in vitro experiments, it could not be sufficient to support the use of terms as gut microbiota. It is essential to re-mark about the wide differences and, therefore, this are bacterial pools, fecal communities, and fecal microbiota and so on. This suggestion concerns all the text, since abstract to conclusion. 

Moreover, even if adopted databases to hierarchically classified raw sequences to OTU need a little more time to include the novel taxonomy, it is more than a year that is not more microbiologically admissible to mention Lactobacillus plantarum. The same suggestion covers all used terms to describe species of lactobacilli, when these are known. Please, revise taxonomy of lactobacilli, supporting novel genera in this time of transition (PMID: 32293557).

Following the same field, please note that not mention was found in methods to describe the next-generation microbiome bioinformatics platform used to assigned sequences as well as no mention was found about the taxonomic databases used to perform taxonomic classification.

Section 3.7.2. OTUs analysis. Can authors explain percentages reported in lines 409-412? "After incubation for 4 h, the relative abundance of Firmicutes increased by 42.14%......in the EPS group". Looking the Figure 9A, it is possible to collected as the relative abundance of Firmicutes in EPS4 is near to the 65-66%. The same is found to all other percentages included within the afore-reported lines. Please, revise.

Another personal suggestion is to separately describe the results and the discussion even if this is not mandatory in accordance to guidelines given by this journal. Regardless the choice to follow the previous suggestion, the results collected after the in vitro fermentations are poorly discussed. In both sections ("3.7.2. OTUs analysis" and "3.7.3. Fecal concentrations of SCFA") the discussion turn around Firmicutes and Proteobacteria (and relative subtaxa, i.e., lactobacilli, Escherischia-Shigella) by showing all fields either pathologies in which these phyla (and related subtaxa) are increased/decreased (LL 413-419). A short mention was found about Actinobacteria in order to give a reference about beneficial effects of Bifidobacterium (LL 438-439)Are the other bacterial taxa that significantly moved of interest? Can authors describe the contribute of EPS to influence the entire microbial community?

With the sentence "Furthermore, L. acidophilus[77] and B. lactis[78] could alleviate the symptoms of allergic nasal symptoms", can the authors speculate about an improvement of "allergic nasal symptoms" as positive outcomes of the in vitro experiments? 

LL442-445. Again evidence about Escherichia-Shigella negatively associated to health.

L.447 What is the threshold given by authors to define the "dominant microflora"? So, if it is, which OTUs are sub-dominant? What about them?

From line 449 to 467, it started a list of all genera and all the related taxonomy (phylum, order, class, family). Taking into account this redundancy, would authors discuss them being these among the significant results?  

L474 "However, the concentration of the total SCFAs during the in vitro fecal fermentation was not significantly changed probably due to the fact that the SCFAs products of some bacteria might provide substrates for others". Ok, if it is, how authors justify the fact that the cross-feeding was not observed in controls? What is the reason for which SCFA (both as singular compounds as well as total) are higher in controls after 4h of fermentation?

Please, italic is required since family to more in depth taxonomic levels (LL 432-438).

L199. 2.9.3.16. S rRNA analysis > 2.9.3. 16S rRNA analysis.

L395. 3.7.1α-. Diversity analysis > 3.7.1- α-diversity analysis.

L434. Change with full stop before "In vitro [...]".

L474. Change with full stop before "However, [...]".

Author Response

Reviewer 1

General comment:The study "Characterization and in vitro intestinal microflora regulatory activity of a low-molecular-weight exopolysaccharide produced by Lactobacillus plantarum NMGL2" purposed by Yao and co-workers aimed to characterize by the application of multiple analytic methods the novel low-molecular-weight exopolysaccharide (EPS) produced by L. plantarum NMGL2. The scientific soundness of this work is of relevance in accordance with the perspective application of this EPS as prebiotic source.

Response: Thanks for the comments.

Major concern:

Comment: The first sections, ones related to chemical and structural characterization of this metabolite, did not require further revisions.

Response: Thanks a lot for the comments.

Comment: Oppositely, the microbiological field of investigation need to be significantly revised and discussed. Berg and co-authors have recently revised the correct terminology to be applied (PMID: 32605663). Therefore, all flora or microflora terms used along the text need to be changed in microbiota accordance to the use of appropriate terminology. Applying an in vitro experiments, it could not be sufficient to support the use of terms as gut microbiota. It is essential to re-mark about the wide differences and, therefore, this are bacterial pools, fecal communities, and fecal microbiota and so on. This suggestion concerns all the text, since abstract to conclusion.

Response: Thanks for your advice. We have modified all the term “intestinal microflora” to fecal microbiota in the manuscript.

Comment: Moreover, even if adopted databases to hierarchically classified raw sequences to OTU need a little more time to include the novel taxonomy, it is more than a year that is not more microbiologically admissible to mention Lactobacillus plantarum. The same suggestion covers all used terms to describe species of lactobacilli, when these are known. Please, revise taxonomy of lactobacilli, supporting novel genera in this time of transition (PMID: 32293557).

Response: Thanks for your advice. We have changed Lactobacillus plantarum to Lactiplantibacillus plantarum, and all other species of lactobailli to new terms in the whole text.

Comment: Following the same field, please note that not mention was found in methods to describe the next-generation microbiome bioinformatics platform used to assigned sequences as well as no mention was found about the taxonomic databases used to perform taxonomic classification.

Response: Thanks for your advice. We have added the microbiome bioinformatics platform and taxonomic database to the section 2.9.3. (LL 205-209)

Comment: Section 3.7.2. OTUs analysis. Can authors explain percentages reported in lines 409-412? "After incubation for 4 h, the relative abundance of Firmicutes increased by 42.14%......in the EPS group". Looking the Figure 9A, it is possible to collected as the relative abundance of Firmicutes in EPS4 is near to the 65-66%. The same is found to all other percentages included within the afore-reported lines. Please, revise.

Response: Thanks a lot for the comments. The relative abundance of the fecal microbiota at the phylum leval is shown in the following table. 42.14% means the increased percentage of the relative abundance of Firmicutes in the EPS4 (after 4 h incubation) group compared with the control4 (after 4 h incubation) group, This was calculated as (0.694878-0.48884)/0.48884=42.14%. All other percentages increased or decreased as shown in the text were calculated in the same way.

For clarity, the figure legend of Fig. 9 was modified as: “Figure 9. Comparison of fecal microbiota as affected by the EPS produced by Lpb. plantarum NMGL2 at phylum level (A) and genus level (B) after incubation of the samples for 4 h (Control4 and EPS4 groups) and 12 h (control12 and EPS12 groups) ”.

Control4

EPS4

Increased or decreased percentage

Control12

EPS12

Increased or decreased percentage

Firmicutes

0.48884

0.694878

+42.14%

0.376987

0.35971

-4.58%

Actinobacteriota

0.257556

0.251215

-2.46%

0.089505

0.147223

+64.48%

Bacteroidota

0.249123

0.049815

-80%

0.254729

0.397389

+56%

Proteobacteria

0.001573

0.002352

+49.48%

0.274428

0.09317

-66.05%

Comment: Another personal suggestion is to separately describe the results and the discussion even if this is not mandatory in accordance to guidelines given by this journal. Regardless the choice to follow the previous suggestion, the results collected after the in vitro fermentations are poorly discussed. In both sections ("3.7.2. OTUs analysis" and "3.7.3. Fecal concentrations of SCFA") the discussion turn around Firmicutes and Proteobacteria (and relative subtaxa, i.e., lactobacilli, Escherischia-Shigella) by showing all fields either pathologies in which these phyla (and related subtaxa) are increased/decreased (LL 413-419). A short mention was found about Actinobacteria in order to give a reference about beneficial effects of Bifidobacterium (LL 438-439)Are the other bacterial taxa that significantly moved of interest? Can authors describe the contribute of EPS to influence the entire microbial community?

Response: Thanks for your constructive suggestion. Describing the results and discussion separately makes the structure clearer. However, considering the structure of the entire article and the fact that the journal does not require the two to be separated, we decided to describe the results and the discussion together. And we have added the description of the effect of EPS on the entire microbial.

Comment: With the sentence "Furthermore, L. acidophilus[77] and B. lactis[78] could alleviate the symptoms of allergic nasal symptoms", can the authors speculate about an improvement of "allergic nasal symptoms" as positive outcomes of the in vitro experiments?

Response: Thanks for your constructive comments. Alleviating the symptoms of allergic nasal symptoms is not a direct positive result of the vitro experiments. After incubation with the EPS, the relative abundance of Bifidobacterium and Lactobacillus increased significantly. However the variation in abundance of L. acidophilus and B. lactis is not clear. To avoid misunderstanding, we have removed the part of L. acidophilus and B. lactis.

Comment: LL442-445. Again evidence about Escherichia-Shigella negatively associated to health.

Response: Thanks for your comment. After incubation with the EPS for 12 h, the

relative abundance of Escherichia-Shigella decreased significantly. However, this could only indicate changes in the abundance of Escherichia-Shigella in fecal microbiota, not the direct effect on human health. To avoid misunderstanding, we have removed the evidence about Escherichia-Shigella negatively associated to health.

Comment: L.447 What is the threshold given by authors to define the "dominant microflora"? So, if it is, which OTUs are sub-dominant? What about them?

Response: Thanks for your constructive suggestion. Referring to Li's article, LDA values more than 4 were used as a threshold to evaluate the dominant microflora. We have added the threshold in this part. The LEfSe analysis was used to evaluate the dominant flora and find out biomarkers between groups after EPS intervention. Therefore we do not focus on the sub-dominant flora(LDA<4).

Comment: From line 449 to 467, it started a list of all genera and all the related taxonomy (phylum, order, class, family). Taking into account this redundancy, would authors discuss them being these among the significant results?

Response: Thanks for your constructive question. We have removed redundant information and kept only the important conclusions.

Comment: L474 "However, the concentration of the total SCFAs during the in vitro fecal fermentation was not significantly changed probably due to the fact that the SCFAs products of some bacteria might provide substrates for others". Ok, if it is, how authors justify the fact that the cross-feeding was not observed in controls? What is the reason for which SCFA (both as singular compounds as well as total) are higher in controls after 4h of fermentation?

Response: Thanks for your question. In vitro fecal fermentation with the EPS for 4 h, there was a significant increase in SCFAs producing genera in the EPS4 group. For instance, the abundance of Phascolarctobacterium and Faecalibacterium increased significantly (p<0.001).

 However, there was no increase in SCFAs. We speculate that the SCFAs were utilized by the Firmicutes, promoting a significant increase in the Firmicutes.

We have added the explanation to the manuscript.(LL458-461)

Specific comments:

Comment: Please, italic is required since family to more in depth taxonomic levels (LL 432-438).

Response: Thanks for your advice. We have corrected all.

Comment: L199. 2.9.3.16. S rRNA analysis > 2.9.3. 16S rRNA analysis.

Response: Thanks for your advice. We have corrected it. We have corrected it with a few adjustments.

Comment: L395. 3.7.1α-. Diversity analysis > 3.7.1- α-diversity analysis.

Response: Thanks for your constructive advice.

Comment: L434. Change with full stop before "In vitro [...]".

Response: Thanks for your advice. We have corrected it.

Comment: L474. Change with full stop before "However, [...]".

Response: Thanks for your advice. We have corrected it.

Reviewer 2 Report

The manuscript foods-1527598 entitled "Characterization and in vitro intestinal microflora regulatory activity of a low-molecular weight exopolysaccharide produced by Lactobacillus plantarum NMGL2" has described isolation, purification, and characterization of exopolysaccharide (EPS). The authors proposed that the EPS could be served as a bioactive agent for potential application in food and health care industries without any specification. The authors need to significantly show the highlight of their work such as prebiotics, dietary fiber, bioactive compounds, and so on. This will guide the authors to the relevant experiments and applications. Low MW EPS has also been reported especially those produced by lactic acid bacteria. With regard to the experimental design, I guess the authors tend to use the EPS as prebiotics. To be used the EPS as a prebiotic substance, it must resist gastrointestinal tract conditions and it must be fermentable by the intestinal microbiota. Therefore, in vitro digestibility test of the EPS is necessary prior to evaluating its fermentability. If the EPS is indigestible under GIT conditions, it will be moved to large intestine where it can be served as food for microbiota. In general, any prebiotics could be fermented by gut microbiota including probiotics, thus providing intestinal environments that promote human health. Fermentation of prebiotics can enhance growth of probiotic, consequently the production of metabolites such as lactic acid and other short chain fatty acids.

In part of intestinal microflora regulatory activity of EPS, I agree with the technique that the authors used for evaluation. However, the results and discussion are unclear to me. My comments and suggestions are as follows.

I recommend adding more experiments to show that EPS is resistant toward gastrointestinal conditions in order to claim that the EPS can be used as a prebiotic substance.

It would be clear if the authors show profile of EPS concentration and pH during the fermentation. This will confirm that EPS is fermentable.

Please provide Fig. 9 more clearly and some legends are unreadable.

Line 405-469, please explain the relation of the relative abundance in terms of phylum and genus level. After comparison, the authors will find that changes of Bifidobacterium are related to Actinobacteria. Changes of Escherichia-Shigella are related to Proteobacteria. These changes directly affect profile of short chain fatty acids.

Since there is no carbon source in the controls, the results showed that short chain fatty acids were produced. How was it possible?

It is well-known that lactic acid is one of the main organic acids produced during fermentation of carbon source by gut microbiota. As stated in Line 436, there was an increase in Lactobacillus sp. Was there any lactic acid produced during the fermentation?

Please give possible pathways where these short chain fatty acids were produced. What kind of microorganisms play important role in production of acetic acid, propionic acid and butyric acid?

Line 500-502, I do not agree with the authors, if there is no evidence to state that the EPS is indigestible after treating with GIT conditions.

Other comments,

Recently, classification of genus Lactobacillus has been published. Please consider using the new name of some Lactobacillus sp.

Line 14, please italicize Lactobacillus plantarum and check that all scientific names of microorganisms are in correct format.

Line 168, please edit the word

Please express xg for centrifugal force. Represent g as the centrifugal force is confusing with g for mass.

Please express min for minute or minutes

Please express number as comma as thousand separators e.g., 10,000

Line 199, 16S rRNA analysis should be changed to extraction of genomic DNA and amplicon sequencing of 16S rRNA gene and bioinformatic tools used to analyze should be added.

As stated in line 104, the linear elution was performed in DEAE-Sepharose fast flow, however, the elution profile present in Fig 1A is a stepwise gradient. Please clarify.

Line 302, please give an example for food application of the EPS.

Line 315, please edit mg ml-1

Line 312-327, based on the results, what is the benefit of using EPS in food application. Please add this information in the manuscript.

Fig 6., please adjust the x axis of Fig. 6B to 24 min

There are too many references (82 references). Please consider presenting only the most important.

Author Response

Reviewer 2

General comment: The manuscript foods-1527598 entitled "Characterization and in vitro intestinal microflora regulatory activity of a low-molecular weight exopolysaccharide produced by Lactobacillus plantarum NMGL2" has described isolation, purification, and characterization of exopolysaccharide (EPS). The authors proposed that the EPS could be served as a bioactive agent for potential application in food and health care industries without any specification. The authors need to significantly show the highlight of their work such as prebiotics, dietary fiber, bioactive compounds, and so on. This will guide the authors to the relevant experiments and applications. Low MW EPS has also been reported especially those produced by lactic acid bacteria. With regard to the experimental design, I guess the authors tend to use the EPS as prebiotics. To be used the EPS as a prebiotic substance, it must resist gastrointestinal tract conditions and it must be fermentable by the intestinal microbiota. Therefore, in vitro digestibility test of the EPS is necessary prior to evaluating its fermentability. If the EPS is indigestible under GIT conditions, it will be moved to large intestine where it can be served as food for microbiota. In general, any prebiotics could be fermented by gut microbiota including probiotics, thus providing intestinal environments that promote human health. Fermentation of prebiotics can enhance growth of probiotic, consequently the production of metabolites such as lactic acid and other short chain fatty acids.

Response: Thans for your constructive comment. EPS from lactic acid bacteria are an alternative as novel potential prebiotics that regulate microbiota. This study was only a preliminary investigation of the effect of EPS produced by L .plantarum NMGL2 on fecal microbiota in vitro fermentation. Therefore, in vitro digestibility test of the EPS was not performed. We will study the prebiotic properties of EPS at the in vivo level in the future, and will conduct in vitro digestibility test in advance. In this study, we have modified "prebiotic activity" to "bioactivity" in view of the fact that in vitro digestibility test of the EPS was not performed.

Major concern:

Comment: In part of intestinal microflora regulatory activity of EPS, I agree with the technique that the authors used for evaluation. However, the results and discussion are unclear to me. My comments and suggestions are as follows.

I recommend adding more experiments to show that EPS is resistant toward gastrointestinal conditions in order to claim that the EPS can be used as a prebiotic substance.

Response: Thanks a lot for the suggestion. As stated above, the EPS would not be claimed as a prebiotic in this study, and in vitro digestibility test will be done in another study later

Comment: It would be clear if the authors show profile of EPS concentration and pH during the fermentation. This will confirm that EPS is fermentable.

Response: Thanks for the comment. In this study, mainly the effect of the EPS on the fecal microbiota was studied, and how the EPS was fermented was not further investigated. In addition, the concentration of the EPS in the fecal sample was difficult to determine due to complex composition of the fecal sample, e.g. possible presence of undigested polysaccharides.  

Comment: Please provide Fig. 9 more clearly and some legends are unreadable.

Response: Thank you for pointing this out. To make the figure clearer we have splitted the original Fig. 9 into Fig. 9 and Fig. 10. In addition, the figure legend of Fig. 9 was modified as: “Figure 9. Comparison of fecal microbiota as affected by the EPS produced by Lpb. plantarum NMGL2 at phylum level (A) and genus level (B) after incubation of the samples for 4 h (Control4 and EPS4 groups) and 12 h (control12 and EPS12 groups) ”

Comment: Line 405-469, please explain the relation of the relative abundance in terms of phylum and genus level. After comparison, the authors will find that changes of Bifidobacterium are related to Actinobacteria. Changes of Escherichia-Shigella are related to Proteobacteria. These changes directly affect profile of short chain fatty acids.

Response: Thanks for the constructive suggestion. We have added more explaination regarding the relevance of change of different phyla or genera, specially in relation to change of SCFAs (Line 426-439).

Comment: Since there is no carbon source in the controls, the results showed that short chain fatty acids were produced. How was it possible?

Response: Thank you for the comment. SCFAs were produced in the control group without adding carbon source, which might be due to the fecal microbiota utilizing the undigested carbohydrates in the feces. This is consistent with the results of Bengoa et al. [1] and Devi et al's [2] studies that the control group also produced short-chain fatty acids in the in vitro fermentation.

[1] Bengoa A A ,  Dardis C ,  Gagliarini N , et al. Exopolysaccharides From Lactobacillus paracasei Isolated From Kefir as Potential Bioactive Compounds for Microbiota Modulation[J]. Frontiers in Microbiology, 2020, 11.

[2] Devi P B , Kavitake D , Jayamanohar J , et al. Preferential growth stimulation of probiotic bacteria by galactan exopolysaccharide from Weissella confusa KR780676[J]. Food Research International, 2021:110333.

Comment: It is well-known that lactic acid is one of the main organic acids produced during fermentation of carbon source by gut microbiota. As stated in Line 436, there was an increase in Lactobacillus sp. Was there any lactic acid produced during the fermentation?

Response: Thanks for the question. Lactobacillus sp. are capable of producing lactic acid from a carbon source. Unfortunately, we did not measure lactic acid production during fermentation. However, Tang et al. [3] found lactic acid produced during EPS fermentation in vitro. Therefore, we speculate that lactic acid may have been produced during the fermentation process in this study.

[3] Tang W , Zhou J , Xu Q , et al. In vitro digestion and fermentation of released exopolysaccharides (r-EPS) from Lactobacillus delbrueckii ssp. bulgaricus SRFM-1[J]. Carbohydrate Polymers, 2019, 230:115593.

Comment: Please give possible pathways where these short chain fatty acids were produced. What kind of microorganisms play important role in production of acetic acid, propionic acid and butyric acid?

Response: Thanks for your advice. We have added this part (Line 426-435).

Comment: Line 500-502, I do not agree with the authors, if there is no evidence to state that the EPS is indigestible after treating with GIT conditions.

Response: Thanks for the suggestion. We agree and have changed the expression "prebiotic activity" to "bioactivity".

Other comments:

Comment: Recently, classification of genus Lactobacillus has been published. Please consider using the new name of some Lactobacillus sp.

Response: Thanks for your advice. We hace changed Lactobacillus plantarum to Lactiplantibacillus plantarum, and also other Lactobacillus sp. to new names.

Comment: Line 14, please italicize Lactobacillus plantarum and check that all scientific names of microorganisms are in correct format.

Response: Thanks for your advice. We have corrected all.

Comment: Line 168, please edit the word

Response: Thanks for your advice. We have edited the complete word.

Comment: Please express xg for centrifugal force. Represent g as the centrifugal force is confusing with g for mass.

Response: Thanks for your constructive suggestion. We have modified all “g” to “×g” for centrifugal force.

Comment: Please express min for minute or minutes

Response: Thanks for your advice. We have modified all.

Comment: Please express number as comma as thousand separators e.g., 10,000

Response: Thanks for your advice. We have corrected all.

Comment: Line 199, 16S rRNA analysis should be changed to extraction of genomic DNA and amplicon sequencing of 16S rRNA gene and bioinformatic tools used to analyze should be added.

Response: Thanks for your constructive suggestion. We have changed “16S rRNA analysis” to “extraction of genomic DNA and amplicon sequencing of 16S rRNA gene” and added the analysis tools.

Comment: As stated in line 104, the linear elution was performed in DEAE-Sepharose fast flow, however, the elution profile present in Fig 1A is a stepwise gradient. Please clarify.

Response: Thanks for pointing out the mistake. We have changed “linear elution” to “stepwise gradient elution”.

Comment: Line 302, please give an example for food application of the EPS.

Response: Thanks for your advice. We have added the examples for food application of the EPS.

Comment: Line 315, please edit mg ml-1

Response: Thanks for your advice. We have modified “mg ml-1” to “mg/mL”.

Comment: Line 312-327, based on the results, what is the benefit of using EPS in food application. Please add this information in the manuscript.

Response: Thanks for your constructive suggestion. We have added the benefit of using EPS in food application in the last part of Section 3.5.1.

Comment: Fig 6., please adjust the x axis of Fig. 6B to 24 min

Response: Thank you for your advice. We have adjusted the x axis of Fig. 6B to 24 min.

Comment: There are too many references (82 references). Please consider presenting only the most important.

Response: Thanks for your constructive comments. We have removed the less important references.

Round 2

Reviewer 1 Report

Into the revised version of the manuscript "Characterization and in vitro fecal microbiota regulatory activity of a low-molecular-weight exopolysaccharide produced by Lactiplantibacillus plantarum NMGL2", authors had addressed the majority of suggestions and comments. 

Some refuses have been found into this version (eg, L348> the subsection was 3.6.2. instead of 3.6.21; L443 and L482> the term flora was not changed).

Additionally, I really suggest to include:

1. the mention about the decrease of butyrate in both batches with EPS (4 and 12 h) compared to both controls (4 and 12 h) into the section about SCFA profiling

2. a brief paragraph onto strengths and limitations of this study before the conclusions

Author Response

We feel great thanks for your professional review work on our article. According to your suggestions, we have made corrections to our previous manuscript, and the detailed corrections are listed below.

General comment: Into the revised version of the manuscript "Characterization and in vitro fecal microbiota regulatory activity of a low-molecular-weight exopolysaccharide produced by Lactiplantibacillus plantarum NMGL2", authors had addressed the majority of suggestions and comments.

Comment:Some refuses have been found into this version (eg, L348> the subsection was 3.6.2. instead of 3.6.21; L443 and L482> the term flora was not changed).

Response: Thanks for pointing out the mistake. We have changed “3.6.21” to “3.6.2” and “flora” to “fecal microbiota”.

Comment:Additionally, I really suggest to include:

  1. the mention about the decrease of butyrate in both batches with EPS (4 and 12 h) compared to both controls (4 and 12 h) into the section about SCFA profiling
  2. a brief paragraph onto strengths and limitations of this study before the conclusions

Response: Thanks for your constructive suggestion. We have added the describetion about the decrease of butyrate.(L509-511) and the strengths and limitations of this study(L523-526).

Reviewer 2 Report

The manuscript foods-1527598R1 have been edited and modified to respond my comments and suggestion. However, the major questions must be clarified: 

  1. With regard to the authors’ response “The study was only a preliminary investigation of the effect of EPS produced by L. plantarum NMG2 on fecal microbiota in vitro fermentation” and “mainly the effect of the EPS on the fecal microbiota was studied, and how the EPS was fermented was not further investigated”

I do not understand the concept of this manuscript. Whether the authors will consider EPS as a prebiotic or a bioactive ingredient, the EPS is being targeted at large intestine where it will be further fermented by microbiota. It is meaningless if the EPS does not resist GIT conditions.

It is unacceptable to state that “The study was only a preliminary investigation of the effect of EPS produced by L. plantarum NMG2 on fecal microbiota in vitro fermentation”. As Foods is Q2 Journal, significant results must be presented. Therefore, in order to show that the EPS was fermentable which is the key result of this study, profile of EPS fermentation must be added in the manuscript.

In fact, profile of EPS fermentation is relative to production of short chain fatty acids. The authors initiated the fermentation using 20 mg/mL of EPS with the prepared fecal slurry of 15% (v/v). This could not be reasons to state the difficulty in determination of total polysaccharides. The fecal sample was diluted with buffer before use as inoculum.  This may not be a reason to avoid profile of EPS. There are many simple methods to determine total carbohydrate such as phenol-sulfuric acid, and Anthrone.

  1. With regard to the authors ‘response “Unfortunately, we did not measure lactic acid production during fermentation. However, ………. Therefore, we speculate that lactic acid may have been produced during the fermentation process in this study”

This is the reason why the authors must present profile of lactic acid production in the manuscript. According to the results of fecal microbiota regulatory activity of EPS, SCFAs were mainly produced by not only Firmicutes but also Actinobacteria. I agree only a part of discussion that the authors provided. Lactobacillus sp. is a member of Firmicutes, and it mainly produces lactic acid and acetic acid. In addition, the results revealed that the majority of Actinobacteria is Bifidobacterium sp. (Fig. 9). Typically, Bifidobacterium produces both lactic acid and acetic acid. Therefore, profile of lactic acid production should unavoidable. If there is no information of lactic acid production, Fig. 11 may not be reliable as the total SCFA obtained from culture supplemented with EPS was not significantly different. It means that EPS fermentation has less effect than undigested polysaccharides. SCFAs are mixed growth associated products, thus being produced whenever growth of the producers detected. Here, lactic acid production from Firmicute and Actinobacteria may play important role in regulation of Proteobacteria especially Escherichia-Shigella as usually found in several studies.

Minor questions:

Line 454: please edit the subtitle to “SCFAs produced in the fecal cultures”

Line 455-470: Please indicate the purpose of determination of SCFA

Author Response

Thanks a lot for your professional review work. We have made corrections to our previous manuscript based on your suggestions, and the detailed corrections are listed below.  

General comment:

The manuscript foods-1527598R1 have been edited and modified to respond my comments and suggestion. However, the major questions must be clarified:

Comment 1:With regard to the authors’ response “The study was only a preliminary investigation of the effect of EPS produced by L. plantarum NMG2 on fecal microbiota in vitro fermentation” and “mainly the effect of the EPS on the fecal microbiota was studied, and how the EPS was fermented was not further investigated”

I do not understand the concept of this manuscript. Whether the authors will consider EPS as a prebiotic or a bioactive ingredient, the EPS is being targeted at large intestine where it will be further fermented by microbiota. It is meaningless if the EPS does not resist GIT conditions.

It is unacceptable to state that “The study was only a preliminary investigation of the effect of EPS produced by L. plantarum NMG2 on fecal microbiota in vitro fermentation”. As Foods is Q2 Journal, significant results must be presented. Therefore, in order to show that the EPS was fermentable which is the key result of this study, profile of EPS fermentation must be added in the manuscript.

In fact, profile of EPS fermentation is relative to production of short chain fatty acids. The authors initiated the fermentation using 20 mg/mL of EPS with the prepared fecal slurry of 15% (v/v). This could not be reasons to state the difficulty in determination of total polysaccharides. The fecal sample was diluted with buffer before use as inoculum. This may not be a reason to avoid profile of EPS. There are many simple methods to determine total carbohydrate such as phenol-sulfuric acid, and Anthrone.

Response: Thanks a lot for your critical suggestion. We mean that the study was carried out to invstigate the effect of EPS produced by Lpb. plantarum NMG2 on fecal microbiota in vitro fermentation. To test whether the EPS resists GIT conditions, we have done the experiment and proved the resistance of Lpb. plantarum NMGL2 EPS to simulated human digestive juice, and this part was added to the manuscript in 2.9 and 3.7. It is true that the total carbohydrate can be determined by the phenol-sulfuric acid method.

Comment 2:With regard to the authors ‘response “Unfortunately, we did not measure lactic acid production during fermentation. However, Therefore, we speculate that lactic acid may have been produced during the fermentation process in this study”

This is the reason why the authors must present profile of lactic acid production in the manuscript. According to the results of fecal microbiota regulatory activity of EPS, SCFAs were mainly produced by not only Firmicutes but also Actinobacteria. I agree only a part of discussion that the authors provided. Lactobacillus sp. is a member of Firmicutes, and it mainly produces lactic acid and acetic acid. In addition, the results revealed that the majority of Actinobacteria is Bifidobacterium sp. (Fig. 9). Typically, Bifidobacterium produces both lactic acid and acetic acid. Therefore, profile of lactic acid production should unavoidable. If there is no information of lactic acid production, Fig. 11 may not be reliable as the total SCFA obtained from culture supplemented with EPS was not significantly different. It means that EPS fermentation has less effect than undigested polysaccharides. SCFAs are mixed growth associated products, thus being produced whenever growth of the producers detected. Here, lactic acid production from Firmicute and Actinobacteria may play important role in regulation of Proteobacteria especially Escherichia-Shigella as usually found in several studies.

Response: Thank you for pointing it out. We agree with you.During in vitro fermentation, SCFAs and lactic acid produced by Firmicutes and Actinobacteria regulate the pH of the fecal microbiota and fermentation system. Therefore, the profile of lactic acid during fermentation should be measured. However, due to objective limitations, the the profile of lactic acid cannot be measured. Considering the actual situation, we have added a description of lactic acid to the manuscript.(Line496-498 and Line512-513).

Minor questions:

Comment 1:Line 454: please edit the subtitle to “SCFAs produced in the fecal cultures”

Response: Thanks for your advice. We have changed it.

Comment 2:Line 455-470: Please indicate the purpose of determination of SCFA

Response: Thanks for your advice. We have added the purpose of determination of SCFA in 3.8.3(Line492-496).